# Investigating a Method for a Horizontal Comprehensive Eco-Compensation Standard of Interregional Ecological Regulating Services

**DOI:** 10.3390/e25091319

**Published:** 2023-09-09

**Authors:** Yejing Zhou, Jingxuan Zhou, Meng Xia, Le Zhang

**Affiliations:** 1School of Civil Engineering and Architecture, Wuhan Institute of Technology, Wuhan 430073, China; 2School of Environmental Science & Engineering, Huazhong University of Science and Technology, Wuhan 430074, China; zjxlypyj@163.com; 3Wuhan Spatial Planning Vetting Center, Wuhan 430010, China; summerhxia@163.com; 4School of Aerospace Engineering, Huazhong University of Science and Technology, Wuhan 430074, China; z18153883147@163.com

**Keywords:** eco-compensation standard, horizontal comprehensive compensation, maximum entropy production, regulating service values, value curved-surface model

## Abstract

Horizontal eco-compensation lacks effective solutions for implementing comprehensive multifactor compensation and determining the compensation standard. To meet those needs, a method named entropy flat surface was proposed and put into practice. However, some scientific problems were found. More specifically, the measurement method of the ecological value was controversial, and the value diffusion model did not reflect the change in the value gradient caused by spatial distance, while the value diffusion had an unclear scope. Therefore, this work optimized and studied the entropy curved-surface method in the case of E’zhou City. The main goal was to build a value-surface model of ecological regulating services based on maximum entropy production. As far as a tangible normal distribution surface is concerned, the model was more consistent with the ecosystem’s energy flow characteristics. The external output of value could be precisely expressed by the dynamic and stable expansion state of the surface model. Therefore, the eco-compensation relations and results were clear. Theoretically, the E’Cheng and Huarong Districts should pay a total of 114 million CNY to the Liangzi Lake District. Our work provided a new perspective, in terms of finding a rule of the ecological regulating service values of the macro morphological structure, simulating the transmission and diffusion of multiple values in interregional areas, solving the calculation problem of the horizontal ecological comprehensive compensation standard, and clarifying the relations of compensation.

## 1. Introduction

Opinions on Improving the Ecological Protection Compensation Mechanism issued by China’s State Council clearly proposed establishing a diversified eco-compensation mechanism, of which horizontal eco-compensation was a vital part [1,2]. Horizontal eco-compensation is an institutional arrangement that adjusts the interest relations between regions without administrative subordination but with close ecological relations [3]. Two main types of eco-compensation are popular, which are (1) a market operation based on the Coase theorem (the Coase theorem states that if trade in an externality is possible and there are sufficiently low transaction costs, bargaining will lead to a Pareto-efficient outcome regardless of the initial allocation of property. In practice, obstacles to bargaining or poorly defined property rights can prevent Coasian bargaining <https://en.wikipedia.org/wiki/Coase_theorem (accessed on 7 January 2019)>. However, the Coase theorem provides a new way of thinking and a method to solve external problems through a market mechanism. Under this theory, the solution for the eco-compensation problem based on market operations belongs to Coasian PES.); (2) government operation based on Pigouvian theory (Pigouvian theory describes that the solution to correct externalities is for the government to correct the private costs of economic parties through taxation or subsidies <https://en.wikipedia.org/wiki/Pigovian_tax (accessed on 7 January 2019)>. As long as the government takes measures to make the private costs and interests equal to the corresponding social costs and interests, the allocation of resources can reach the optimal Pareto state. Similarly, government-operated eco-compensation belongs to the Pigouvian PES.). The Coasian and pure-market approach generally is regarded as a more reasonable and democratic way of eco-compensation [4]. However, it cannot be easily implemented in practice because of the reluctance of individuals to pay for ecological services [5]. Currently, the majority of eco-compensation practices in the world require the government to push and operate through mandatory policies, taxation, transfer payments or other forms of implementation, which may be considered low flexibility and low efficiency [6,7,8,9].

It is important to explore and pilot horizontal compensation in China. First, it can address the defects of vertical transfer payments, which can promote market-based and diversified development. Second, it can reduce financial expenditure pressures on the central government and motivate local potential. Third, the level of ecosystem services and their provision could be improved by the use of horizontal compensation. Fourth, it helps clarify the relation between the rights and responsibilities of adjacent administrative regions [10,11].

China’s horizontal eco-compensation focuses mainly on the upstream and downstream compensation of river basins [12,13]. In the past decade, some instances of representative horizontal compensation have been implemented, such as the Xin’an River Basin compensation: Zhejiang and Anhui provinces each contributed 100 million CNY, while China’s central government contributed 300 million for water environment management [14]. That amount resulted from consultation between the two provinces under the guidance of the superior (the central government then withdrew from the project while encouraging the provinces to introduce market-oriented capital). Given that success, the Tingjiang, Hanjiang, Jiuzhou, Luanhe, and Chishui Rivers have also implemented cross-provincial horizontal eco-compensation, which has generally achieved good results [15].

Overseas eco-compensation is included in most laws, governmental taxes, and preferential policies closely related to ecosystem protection [16,17]. Specific horizontal eco-compensation is carried out mainly through purchases of ecosystem services directly by local governments or other market economic methods [18,19,20,21]. In general, most foreign scholars emphasize the important role of politics, institutions, and culture in implementing eco-compensation plans, or they focus on the importance of micro-individual differences in compensation design [22,23]. They also attach importance to the influence of compensation plan participants’ decisions on improving eco-compensation [17,18,19].

Generally, horizontal eco-compensation in China has the following problems: (1) most are pilot projects, have a single element, and lack comprehensive interregional compensation. The fields of practice and research are basin and water resource compensation, most of which are paid by users and polluters [10,22]. The object of compensation is water quality and quantity; there is little comprehensive compensation for various ecological regulating services between regions [24]. (2) Most of the compensation is unscientific and subjective [25,26]. Unlike standards in Western countries, few horizontal eco-compensation standards in China can be calculated and implemented according to scientific theory [27]. No interaction occurs between the recipient and the compensator, but higher government directly promotes or guides the negotiation among the stakeholders, and the compensation scope and scale are small [15,28,29]. China has not established a standard horizontal transfer payment system, and the stakeholders’ willingness to compensate horizontally is weak [24,30]. Therefore, full consultation is needed between the compensator and the recipient with the government’s guidance, and incentive mechanisms are needed to explore a sustainable system design [2,31].

With the improvements in China’s environmental management system, eco-compensation follows the principle of “who benefits and who compensates” [1,32,33] and generally develops in a multifactor, comprehensive, and systematic direction. However, due to the externalities of the ecological regulating services, the complicated compensation relations make it difficult to determine the compensation standard [34,35]. Given the output areas of multifunction and multifactor regulating services, scientific solutions to the problems of compensation scope, objects, amounts, and so forth are still lacking and need a breakthrough [15,36].

Because E’zhou City began to explore a needed eco-compensation scheme in 2017, the authors of this study designed and studied a compensation method for that area. In 2019, a paper was published by us titled “Study on eco-compensation standard for adjacent administrative districts based on the maximum entropy production” [34]. Based on the scientific theory of maximum entropy production (MEP), that method, called entropy flat surface, solved the above problems to a certain extent and provided a standard method for interregional compensation. The research was awarded “Typical Cases of Value Realization of Ecological Products (First batch)” by the Ministry of Natural Resources of China, which was promoted nationwide in 2020 [37].

However, the original method is in its initial stage and must be further developed theoretically. For example, the model of output value cannot reflect the change in value gradient caused by spatial distance, and the scope of value diffusion is unknown. On that basis, with the support of the Hubei provincial government, the provincial department of natural resources took the Liangzi Lake District of E’zhou City as a pilot in the Notice on Pilot Project of Ecological Compensation for Realizing the Value of Ecological Products. That pilot project required continuous optimization of related research work, making up for the existing deficiencies and improving the scientific and systematic methods. The E’zhou municipal government has paid much attention to eco-compensation work and will continue it in the future. Consequently, these researchers chose E’zhou City as the object because it met practical requirements and was significant for deepening their research.

Therefore, the research goal was to find the structural rules of the macroscopic existence of functional values in space and provide a new method for interregional horizontal eco-compensation. In the perspective of methodology, a new concept was associated with the original exploration, which was called an “entropy curved surface”. Particularly, an ecological regulating service value surface was established, combining environmental sciences, ecological economy, mathematical statistics, and other perspectives. In addition, mathematical modeling was implemented, and a geographical information system (GIS), remote sensing, and computer technologies were cross-integrated, and this was applied to the analysis of the public environmental management problems. The method was used to determine the output range and quantity of the ecological regulating service values, and the eco-compensation model was optimized to calculate the compensation standard objectively and to solve the problems of the quantitative evaluation method of the horizontal compensation standard.

## 2. Materials and Methods

### 2.1. Research Ideas

In 1935, A.G. Tansley coined the term “ecosystem” to refer to a comprehensive system consisting of a community of organisms, their abiotic environment, and their dynamic interactions. The whole system should be considered, including biological and physical factors, and these components cannot be separated or treated separately. By proposing that ecosystems are dynamic, interacting systems, Tansley’s concept of ecosystems evolved into modern ecology. Modern ecology has directly attracted people’s attention and research on ecosystem energy flow [38,39]. The underlying idea of this work was to consider the eco-compensation process from the idea of system, whole, and functional flow.

In our opinion, the ecological function of each region not only serves the local area but has fluidity and overflow outward [40,41] and affects such factors as gas regulation, climate regulation, and environmental purification. Horizontal eco-compensation is necessary to consider the calculation and supplementation of the fluidity value. The essence of flowing ecological value is that matter and energy are transported. However, how can the ecosystem values be calculated, and how can the complex multisystem and multifunction compensation problems be analyzed?

Eco-compensation should value the ecosystem first. The value of the ecosystem service can be calculated by the method of “equivalent factor per unit area” [42]. That method was inspired by R. Costanza and first issued by Prof. Xie. It systematizes the measuring of the ecosystem function value and aggregates it into a table (Table A1 in Appendix A). The method has significant advantages in evaluating the overall service function value of an ecosystem: it has consistent standards, is intuitive and easy to use, has fast and comprehensive evaluations, and has fewer data needs and strong generalization. Therefore, it has a good application prospect in planning decision making and policy making [43].

Maximum entropy production theory provides a systematic solution in the face of a multidimensional complex ecosystem. MEP assumes that an open system like the Earth, far from having thermodynamic equilibrium, may form a stable and ordered structure in which energy is dissipated and entropy is produced. The open system maintains a stable non-equilibrium state at the rate of maximum possible entropy increase [44,45]. In other words, the motion of matter and energy on the Earth is in a state of dynamic equilibrium. A variety of ecological regulating service functions in a particular area (such as gas regulation, climate regulation, waste treatment, and water regulation shown in Table 1) and the dynamic flow of matter and energy happens continuously. In a given unit of time, such as a year, the process of the ecological regulating service flows follows a specific statistical rule, and if a given mean and variance were set, it is likely to form a normal distribution [46,47]. That state can be described by an ecological regulating service value surface. The central limit theorem provides theoretical support for the value surface tending to a normal distribution.

Based on MEP, it is conceived that if the value of the ecological regulating services is visualized, the dynamic equilibrium state of the value surface is likely to form a normal bell-shaped distribution. The value curved surface expands around to form a dynamic and stable surface structure [48,49]. When it extends to the adjacent districts, the surface can be calculated to obtain the ecological regulation output value, and the transmission relation of eco-compensation is formed naturally.

### 2.2. Framework

Based on the above novel ideas, the research in this study was under the framework of “calculation, modeling, and validation” to study the eco-compensation methods. Relevant information and data were collected and investigated. The main parts were (1) calculation of ecosystem and regulating service values based on the equivalent-factor method, (2) construction of an ecological regulating value curved-surface model based on MEP, and (3) validation of the curved-surface model. The second part was the focus of this study. Parts 1 and 3 supported the value data in the early stage and the empirical study in the later stage. Finally, the model was used to calculate the output value and determine the compensation standard. Specific study steps can be seen in Figure 1 and in Section 2.3.

### 2.3. Methods

#### 2.3.1. Research Data and Location

E’zhou City is in Hubei Province, China, which comprises the E’cheng, Huarong, and Liangzi Lake Districts. The three districts are interregional and adjacent to each other, covering an area of 1596 km^2^ (Figure 2). Eco-compensation is conducted among the three districts. Demand data include remote sensing images, environmental monitoring data, water conservancy data, meteorological data, the Third National Land Survey data (“third survey” for short), agricultural product prices, and social and economic data within the research area.

#### 2.3.2. Ecosystem Value

1. Correspondence between land type and ecosystem type: According to the ecosystem types of Table 1 [43] and the land data of the third survey, from the concept, the land types of E’zhou city and the ecosystem types in the table were selected and corresponded.

2. Adjustment of the equivalent-factor value: The factor values of different levels were determined in accordance with the water quality and forest stock grades in the Natural Resources Balance Sheet (the Natural Resources Balance Sheet is a natural resources status report reflecting the levels and changes of natural resource assets and their value at the beginning and end of a year [50]). Then, a new equivalent-factor table (Table 1) was adjusted and established.

3. Value of ecosystem (ecosystem value is an important part of natural resource assets. It includes ecosystem stock values comprising forest, wetland, ocean, and other ecosystems, and ecological resources such as water and biological, marine, environmental, and other resources. Ecosystem value also includes the ecosystem flow value, consisting of ecosystem products produced by the ecosystem, environmental regulation, water regulation, and other ecosystem service values [51]) and their regulating services: Using Table 1, the values of a standard equivalent factor (according to the comprehensively determined prices of agricultural products including rice, wheat, and corn), the resources’ areas and the ecosystem were calculated using Equation (1) in the study area. Table 1 includes 4 regulating services (gas regulation, climate regulation, waste treatment, and water regulation) that have the characteristics of flow and diffusion. Their concepts can be seen in Table A2. In the geographic space, GIS was used to grid the 4 regulating service values, and the value distribution was recorded as *H*_0_:(1)$$ESV\left(t\right)=\sum_{j=1}^{8}\sum_{i=1}^{11}S(j)\times X(i,j)\times D(t)$$
where *ESV* is the ecological service value, *t* is a particular year, *i* is a specific ecological function (total 11 functions), *j* is an ecosystem type (total 8 types), *S* is the area of the ecosystem type, *X* is the revised equivalent factor, and *D* is the price of a standard equivalent factor whose value follows the Consumer Price Index (CPI).

#### 2.3.3. Ecological Regulating Service Value Curved-Surface Model

1. Position of the ellipse on the projection of the curved surface: The geographic boundary coordinates of each study area were determined using GIS analysis, and the coordinates were imported into MATLAB R2013b for ellipse fitting. The center point of the fitting ellipse was (*x_j_*, *y_j_*), the length of the major axis was *Lx*, and the length of the minor axis was *Ly*.

2. Basic model of a single-value surface: A curved surface will likely express a bell-shaped normal distribution form. A 3D coordinate system *x*-*y*-*z* was established in MATLAB, where *x*-*y* represents geographical coordinates, and *z* represents ecological value per unit area. The integral of the initial value surface to the fitted ellipse area was equal to the total value of the corresponding ecological regulating service in the study area. No matter how the surface shape changed, the total value remained unchanged. The 2D normal probability density function was proposed to construct the curved surface, that is, the potential function of the value flow—the general solution of the differential Equation (2) (set as *f_j_*).
(2)$$f_{j}=a_{j}\cdot exp\left[-\frac{1}{2}\left(\frac{{\left(x-x_{j}\right)}^{2}}{({{Lx}_{j}b)}^{2}}+\frac{{\left(y-y_{j}\right)}^{2}}{{{(Ly}_{j}c)}^{2}}\right)\right]$$
where the value of *a_j_* determines the total size of the ecological value contained in the surface; the sign of *a* is positive to indicate that the surface is convex; *a_j_
*= *H_j_*/[2π (*Lx_j_ b*)(*Ly_j_ c*)]; *H_j_* is the value of an ecological regulating service measured in the *j* region; *Lx_j_* is the length of the long axis; *Ly_j_* is the length of the short axis in the *j* region; and *b* and *c* are the adjustment coefficients, which are used to adjust the flattening degree of the surface in the x and y directions, respectively.

3. Surface expansion state: *f_j_* expands in all directions, and the expanded shape on the plane was consistent with that of all study areas. In MATLAB, the initial state of *f_j_* is that *b* and *c* are 1. Then, parameters *b* and *c* gradually advanced the value to 1, 2, 3, …, *n* in the trial calculation programming operation, achieving a surface shape that gradually became a flat, smooth process.

4. Surface extension boundary setting: The *μ* value was set, which was the base value of the unit area ecological value within the research range, such as wasteland and bare land values. When the functional value was *z* = *μ* on the surface, the range formed by the surface coordinate was the judgment value *B_j_* of the diffusion boundary. The curved surface expanded outward, and the height decreased, and its stability can be considered if the following conditions are met. The trial was programmed to (1) ensure that the surface extended to cover more than 90% of the research land area, and (2) at the same time, the value within the scope of the extended value-containing surface accounted for over 90% of the total value. The surface shape can be accepted if the above conditions are satisfied.

5. Synthesis of multiple surfaces: Multiple surfaces were superimposed using functions (Max {*f*_1_, *f*_2_, …, *f_n_*}) as the envelope surface, and then the synthetic surface *M* was made continuous and smooth using the moving average filtering method. The peripheral boundary of surface *M* was *B*’, which was part of *B_j_*’s peripheral synthesis (see Figure 3a,b).

#### 2.3.4. Model Validation and Calculation of Eco-Compensation

1. Screening of suitable natural physical quantities and validation of the design: Given the regulation function, the physical quantities were selected by combining qualitative and quantitative analysis methods. They included data queries and literature analyses to judge their relation to ecological value on time scales, different seasons, and ecological benefit impacts. Climate regulation may be related to temperature and humidity; concentrations of particulate matter, nitrogen oxide, and volatile organic compounds can reflect gas regulation. Water regulation may be closely related to precipitation and evapotranspiration. Many objects had to be tested, and the relations were complex, so the specific test design had to be flexibly adjusted based on the actual situation and data availability.

2. Distribution surface of natural physical quantity: Measurable physical quantities, such as air pollutants, temperature, and humidity, were monitored multiple times in different time periods using Sniffer4D, a mobile air-monitoring device, to obtain multiple sets of discrete data. A trend surface of physical quantities was fitted by using a spatial interpolation and global polynomial method (the fitting surface was collectively called *Q*). In corresponding geographical coordinates, several batches of data were randomly extracted from value surface *M* and trend surface *Q* to form vectors to be verified.

3. Validation of surface model: The corresponding variables on the *M* and *Q* surfaces were verified by a 2-sample *t*-test and a Pearson correlation analysis, respectively, to test the mean difference and correlation between the 2 data groups. If the *t*-test difference was significant, it indicated that the samples were not comparable or there were problems and the data should be checked from the source, or the test group should be redesigned. If the correlation analysis was weak or opposite, the model had to be revised to modify its spatial position or adjust the surface’s flatness. If *Q* showed prominent non-normal distribution characteristics, other probability density functions could be used to reset the model.

4. Output value of the regulating service and eco-compensation results: For the value output area, the extended surface would cover surrounding areas. By integrating the value of the area, the value obtained in the value input area can be calculated. The output value of each research area can be calculated, and then the expenditure and income of each research area can be compared to calculate the net output value and determine the eco-compensation amount.

## 3. Results

### 3.1. Ecosystem Value in E’Zhou City

#### 3.1.1. Adjustment of the Corresponding Relation between Land and Ecosystem

There are 13 land categories according to the third survey: wetland, cultivated land, planting garden, forest land, grassland, commercial and service land, industrial and mining land, residential land, public use land, special land, transportation land, water and water conservancy facilities land, and other land (“wetland” is the latest addition) (the Rules for the Identification of Land Categories for the Third National Land Survey (provided by the Ministry of Natural Resources of China) shall apply to the latest land classifications. It added “Wetland” in “Current Land Use classification GB/T 21010-2017”, which has the original 12 land categories).

In light of the corresponding relations between land use properties and equivalent factors, and combined with the characteristics of natural resources and the environment in E’zhou, the following eight main systems and land use classifications are proposed:I.Paddy field—corresponds to paddy fields and irrigated land in “cultivated land”II.Dry land—corresponds to the dry land in “cultivated land”III.Broad-leaved forest—corresponds to tree forest, bamboo forest, shrub forest, and other forest typesIV.Garden—corresponds to “planting garden”V.Shrub grass—corresponds to “grassland”; it is mainly general grassland, without natural/artificial forage grass, etc.VI.Water area and wetland—corresponds to “water area and land for water conservancy facilities”, including rivers, lakes, ditches, reservoirs, etc.VII.Desert—corresponds mainly to “other land”, including sandy land, bare land, rock and gravel land, etc.VIII.Bare land—corresponds to all kinds of construction land, such as commercial service, residential, facilities’ agricultural land, industry, mining, storage, roads, special land and idle land, etc.

#### 3.1.2. Adjustment of Equivalent-Factor Table

Based on the original equivalent-factor table (see Table A1), III, IV, and VI in Table 1 are adjusted factors, taking forest quality and water quality into account, while other factors remain unchanged. The specific details are as follows: (1) the medium quality of broad-leaf corresponds to the value of the original table and is reduced by 50% in the original value, corresponding to “low”, and the other is increased by 50%, corresponding to “high.” (2) The garden factor is half of the broad-leaved forest and paddy field factor. Referring to the Technical Provisions for Continuous Inventory of National Forest Resources, the forest is “low” because its average stock is less than 50 m^3^/ha. (3) The factors reflecting water quality are divided into five levels in accordance with the Environmental Quality Standards for Surface Water, among which Class IV corresponds to the original factors (because the surface water of lakes in China is approximately Class IV), and other levels increase or decrease by 20% successively. In addition, the water regulation factors of Class I to V remain unchanged (Table 1).

The economic value corresponding to a standard equivalent-factor value was 3406 CNY/ha in 2010 [43]. From 2011 to 2019, according to the annual average Consumer Price Index (CPI) in China, the economic value of a factor was 4256 CNY/ha. After adjustment, the adjusted table, the land area data of the third survey, and a standard equivalent-factor value were used to calculate the ecosystem value of the three districts. The results show that the ecological value of E’zhou City was 15.63 billion CNY, that of E’cheng District was 5.224 billion CNY, that of Huarong District was 5.123 billion CNY, and that of the Liangzi Lake District was 5.284 billion CNY. The unit area values of the three districts were 85,956 CNY/ha, 103,959 CNY/ha, and 106,548 CNY/ha, respectively. Overall, the Liangzi Lake District was slightly higher than the other two areas.

The values of the regulating service (includeincludeing gas regulation, climate regulation, waste treatment, and water regulation) were obtained: (1) regulating service values in E’cheng District were 298 million, 526 million, 394 million, and 2.114 billion CNY, respectively. The total regulation value was 3.332 billion, accounting for 63.78% of the ecosystem value. (2) For the Huarong District, they were 277 million, 438 million, 381 million, and 2.177 billion CNY, respectively. The total was 3.273 billion, accounting for 63.89%. (3) For the Liangzi Lake District, the values were 291 million, 518 million, 394 million, and 2.124 billion CNY, respectively. The total was 3.327 billion, accounting for 62.96%.

### 3.2. Curved-Surface Model Building

The GIS data of the boundary and grid center point coordinates were imported into MATLAB. The boundary coordinates were used for ellipse fitting and the standard ellipse equation; the grid center point coordinates were used for value integration. The research area was divided into several 2 × 2 km grids, and the value integral was obtained by multiplying the value on the surface at the grid’s center point by the grid’s area.

#### 3.2.1. Initial Form of the Curved Surface

The coordinates of the administrative area boundary of each research area were input into MATLAB, and ellipse fitting was performed. The ellipse boundaries of Liangzi Lake District, Huarong District, and E’cheng District, the center point *O*, the long and short axes *Lx* and *Ly*, and the inclination angle *θ* were obtained, respectively.

The general elliptic Equation (3) can be obtained by elliptic fitting of the boundary of three districts:(3)$$Ax^{2}+Bxy+Cy^{2}+Dx+Ey+1=0$$

Then, the central coordinates, the long and short axes, and the long axis inclination angle of the ellipse can be obtained using Equation (4):(4)$$\begin{array}{l} X_{C}=\frac{BE-2CD}{4AC-B^{2}},Y_{C}=\frac{BD-2AE}{4AC-B^{2}}\\ Lx=\sqrt{\frac{2(AX_{C}{}^{2}+CY_{C}{}^{2}+BX_{C}Y_{C}-1)}{A+C-\sqrt{{\left(A-C\right)}^{2}+B^{2}}}},Ly=\sqrt{\frac{2(AX_{C}{}^{2}+CY_{C}{}^{2}+BX_{C}Y_{C}-1)}{A+C+\sqrt{{\left(A-C\right)}^{2}+B^{2}}}}\\ \theta =\frac{1}{2}\arctan\frac{B}{A-C} \end{array}$$

The results are as follows (subscripts 1, 2, and 3 represent Liangzi Lake, Huarong, and E’cheng Districts, respectively) (Figure 4a):(1)*O*_1_ = (914,938.9, 3,237,419.9), *O*_2_ = (919,735.9, 3,270,589.9), *O*_3_ = (939,920.4, 3,255,522.4);(2)*Lx*_1_ = 18,866.5, *Ly*_1_ = 9845.6, *Lx*_2_ = 14,767.4, *Ly*_2_ = 10,959.7, *Lx*_3_ = 16,910.5, *Ly*_3_ = 10,287.0;(3)*θ*_1_ = 87.8°, *θ*_2_ = 55.5, *θ*_3_ = –10.7°

Figure 4b,c shows the initial value surfaces of Liangzi Lake District, taking gas regulation as an example. Under the initial conditions, the regulation value is mainly concentrated in its own region but not fully diffused to the other two districts (this moment in the model *b* = *c* = 1). The value surface is essentially an infinite outward expansion and has value at the far end of the surface, but the distribution at the far end is very thin. The dark purple represents a low value: the large area in Figure 4 shows the color.

#### 3.2.2. Extension of Curved Surface

The above method shows that *μ* reflects the ecological background value, and the cross between the *μ* value and the surface defines the range and forms the boundary *B_j_*. To meet the requirements of 90% coverage of the study area and the containing value of not less than 90%, the selection of the background value requires trials according to the actual situation, so the specific selection of the *μ* value set three scenarios: (1) paddy field, dry land, bare land, wasteland; (2) dry land, bare land, wasteland; (3) bare land, wasteland.

After a trial calculation, the *μ* was selected as the third case; paddy field and dry land belong to farmland, so, as the background value, their value is relatively high, therefore only the bare land and wasteland value was considered as the background *μ* value.

Removing the “highland” area of ecological value (including farmland, forest, grassland, water, and wetland), the base value (bare land and wasteland) *μ* of the four regulations were 2115.9, 777.1, 8714.6, and 3523.6 CNY/km^2^, respectively.

If the above conditions are met, the shape of surface diffusion is a stable state and a high-probability event with high reliability. After calculation, the adjustment coefficients *b* and *c* of the ecological value-surface diffusion to stability are shown in Table 2.

#### 3.2.3. Eco-Compensation Results

In this case, E’cheng pays Liangzi Lake and Huarong 151.97 million and 132.37 million CNY, respectively; Liangzi Lake District pays E’cheng and Huarong 107.22 million and 128.30 million CNY, respectively; and Huarong District pays E’cheng and Liangzi Lake 132.62 million and 197.95 million CNY, respectively. The final net payment is that the Liangzi Lake District gains 44.50 million from E’cheng and 69.90 million from Huarong, totaling 114.40 million CNY. The mutual payments among the three districts are shown in Figure 5 and Table 3.

The ecological value surfaces of the three districts of E’zhou were superimposed to obtain the synthetic ecological regulation value-curved surfaces, as shown in Figure 6.

### 3.3. Validation of Curved-Surface Model

Two validations were conducted in total. The data provided by government departments for the first test were of limited variety and covered a small area. The validity of the model was initially discussed from only two perspectives: the relation between the gas regulation and the PM_2.5_ average concentration, and the relation between the waste treatment and the average concentration of ammonia nitrogen in water.

The first validation idea was to compare the data of the regulating service value surface with the monitoring data. In the study area, the higher the gas regulation value, the lower the PM_2.5_ concentration should be; the higher the waste treatment value, the lower the concentration of ammonia nitrogen in the water, showing a negative correlation.

The first validation was unsatisfactory. The correlation between PM_2.5_ and the gas regulation value was 0.84, whereas the correlation between the ammonia nitrogen and the waste treatment value was 0.077, contrary to the verification logic (it should have been a negative correlation). That indicates that the data used had almost no correlation with the value surface.

Considering the problems in the first validation process, the portable air monitor Sniffer4D was adopted to obtain more environmental elements and a more extensive geographical range of data. The equipment records a batch of data every second and monitors the environmental elements by vehicular means (Figure 7). The specific elements include PM_1_, PM_2.5_, PM_10_, SO_2_, NO_2_, O_3_, CO, VOCs, air pressure, humidity, temperature, longitude, and latitude (PM_1_ is ultrafine particulate matter, PM_2.5_ is fine particulate matter, and PM_10_ is inhalable particulate matter. SO_2_ is sulfur dioxide, NO_2_ is nitrogen dioxide, O_3_ is ozone, CO is carbon monoxide, and VOCs are volatile organic compounds. They are common air pollutants.). In the second validation, air humidity and climate regulation, PM_2.5_, PM_10_, VOCs, and gas regulation were initially selected to determine the model’s effectiveness.

The second validation idea was as follows: similar to the first time, the higher the gas regulation value, the lower the PM_2.5_, PM_10_, and VOC concentration. The value and monitoring data should show a negative correlation; the greater the value of climate regulation, the higher the relative humidity, and the value data should show a positive correlation with the monitoring data.

1. Validation of gas regulation value: Several PM_2.5_, PM_10_, and VOC data obtained by the device were randomly selected by GIS. There was no significant difference between the data and gas regulation by *t*-test, *t*_(PM2.5)_ = 0.3133, *t*_(PM10)_ = 0.1874, *t*_(VOCs)_ = 0.5679, all >0.05. Figure 7 shows the data collection track. Those data were used to form a surface through GIS spatial interpolation, which covered the whole area of E’zhou. Then, the values extracted from random points generated on the surface formed the corresponding (*x*, *y*, *z*_1_) values. After that, the parameters of the gas regulation value surface were extracted at the corresponding coordinate points, and the values of (*x*, *y*, *z*_2_) were obtained. *z*_1_ and *z*_2_ were tested for correlation. The correlations between PM_2.5_, PM_10_, VOC data, and gas regulation data directly output by Excel were −0.11, −0.53, and −0.05, respectively (Figure 8a–c).

2. Validation of climate regulation value: There was no significant difference between humidity and climate regulation by *t*-test (*t* = 0.6845, *p* > 0.05). The humidity space was interpolated by GIS to form a surface, and then the values extracted from random points were generated to form the corresponding (*x*, *y*, *z*_1_). The parameters of the climate regulation value surface were also extracted at the same coordinate points, and the values of (*x*, *y*, *z*_2_) were obtained. *z*_1_ and *z*_2_ were tested for correlation. The correlation between the two columns of data directly output by Excel was 0.253. The surface of the climate regulation value and the sample points of the humidity value were drawn using MATLAB (Figure 8d).

From the comparison of natural physical quantities (PM_2.5_, PM_10_, VOCs, and humidity) with the gas regulation and climate regulation surfaces, the verification result was an improvement over the first verification after the data range expansion.

First, in the first validation, the amount of data and the coverage were too small, the comparability was weak, and the test results were illogical. The second validation was an improvement. Although the negative correlations between PM_2.5_ and VOC and gas regulation were insignificant, they were in general accord with the validation logic, and PM_10_ had a good negative correlation. Additionally, the comparisons between different physical quantities and gas regulations did not have ideal results. The result of PM_10_ at least provides evidence that the surface was comparable with some air pollutants. Moreover, climate regulation also showed a positive correlation with humidity.

Second, there was a large conceptual difference between natural physical quantities and the value of the curved surface itself. The natural one was arbitrary and not predictive, while the surface was a conceived ideal value concept derived from the ecological economy and based on the MEP assumptions. Consequently, comparison and validation of the two were a considerable challenge and difficult to test, so they were unlikely to result in a high correlation. Therefore, at least the trend of the above results was ideal, which also provides evidence for the model’s credibility.

## 4. Discussion

### 4.1. Policy Implications

In China, most eco-compensation and its promotion are initiated and implemented by the government. As the comprehensive decision maker, the government not only wants to know the will of the public but also to understand the value of the ecosystem and eco-compensation amount from a macro and systematic perspective. At present, China attaches great importance to environmental protection. Key river basins and localities across the country are promoting and exploring horizontal eco-compensation. The eco-compensation laws or policies follow the principle of “who benefits and who compensates”, and the specific compensation content, standard, and method are usually carried out according to contracts signed between different regions.

Our work has been adopted by the local government to formulate the three-zone compensation policy in E’zhou City. From 2017 to 2019, Liangzi Lake District received eco-compensation of 50.31 million CNY, 82.86 million CNY, and 105.31 million CNY, respectively, jointly paid by E’zhou City Finance, the E’cheng District, and the Huarong District. The Liangzi Lake District of E’zhou City is an ecological environmental protection area, where the regional development is backward, and the per capita income is low after the restriction of development. Eco-compensation in E’zhou City has a positive impact on promoting local development, and the funds can be used to maintain the cost of environmental protection, promote infrastructure construction, motivate green industries or tourism, and improve the income and living standards of local people.

### 4.2. Method Comparisons

Our previous research work published in 2019 can be called the Entropy Flat Surface method (version 1.0). The basic idea and characteristics of version 1.0 were that each research unit’s ecological function (and its value) would spread to the surrounding area. It was assumed that that value diffusion would fully spread in the larger scope, consisting of the research area and the designated area, and would reach a uniform state. In other words, ecological functions and values are not distributed like a bell curve but like a plane, which is an ideal extreme case. That assumption made the mathematical model simple. Only the difference between the unit area of each high ecological value and the ecological average value could be obtained to calculate the total ecological value of the highland flowing out. Then, the compensation standard of each unit in the research area was revised and optimized by considering the actual economic development situation. In the calculation process, three districts having three equivalent-factor tables were used to reflect the environmental quality differences. Fewer data, including the equivalent-factor table, land area, GDP, and so on, were needed for calculation and could be obtained through Excel (Microsoft Office 2019).

The output of the regulating service value in this study was calculated using the entropy curved-surface method (version 2.0); its important characteristics and optimization lie in the following: (1) the eco-compensation method is more systematic and scientific than the previous method. First, the previous method exaggerated the ecological value gap between districts to reduce the subjective adjustments of equivalent factors in calculating value. Second, the value diffusion model is more in line with the description of MEP. The ecological value in highlands expanded outward, and the value distribution was in the form of a normal distribution surface rather than a plane; that was a marked improvement. The ecological value in this model was concentrated in the center, and the further it expanded, the lower the value became, leading to an apparent decrease in the value of ecological output. (The output of version 1.0 was 270 million CNY [36], while in this case, it was 110 million CNY.) It is not that the greater the compensation, the better; we hoped to obtain more scientific calculation results. (2) The eco-compensation process was programmed, standardized, and objectified. In version 1.0, the output value’s diffusion scope, namely, the designated area, and the equivalent factors in three tables contained artificial subjectivity or were predetermined. However, the original factors were not modified very much in this study, which kept the factors consistent and standard for the three districts, resulting in a pronounced decrease in the total amount of compensation. (3) Steps were simplified, and there was a highly optimized logical process. Version 2.0 dispensed with the designated area (it referred to administrative units that surrounded the study area) and simply analyzed the study area. The analysis of GDP and the ecosystem service value (ESV) in the study area and designated area, and the distribution of compensation amount were all removed. At that point, it was no longer necessary to extract the compensation weight determined by GDP and ESV for the compensation results. What the government needs is the most objective result possible. As for some specific operations of eco-compensation, whether or not a district has the ability and willingness to pay, it needs comprehensive coordination.

The previous and new methods are compared in Table 4.

### 4.3. Limitations and Future Work

(1)Methods for value evaluation

The limitation of the equivalent-factor method is that it is considered by some scholars to be subjective and unscientific. There are many ways to account for ecosystem value. The first type is the market- or semi-market-based theory in economic principles. From the perspective of ecological protectors, the standard of eco-compensation is proposed considering direct input and opportunity cost. From the perspective of profit of ecological beneficiaries, the contingent valuation method, the selection test method, the market value method, and the alternative market method are commonly used to measure the basis of payment of beneficiaries [4,20].

Second, from the perspective of ecosystem service value, the value evaluation can be obtained by determining the amount of ecosystem service functional quantity and the unit price of functional quantity, which is referred to as the “Functional value method” (the equivalent-factor method which is suitable for ecosystem valuation in China is derived from it). More specifically, like the market value method, the opportunity cost method, the shadow engineering method, the human capital method, the tourism cost method, the production cost method, etc., are usually used to estimate the value of the ecosystem service function [23]. “Functional value method” is the basic method for carrying out value evaluation and is the mainstream method for determining eco-compensation standards at home and abroad.

In September 2020, the “Technical Guideline on Gross Ecosystem Product” (1.0 Version), i.e., GEP accounting was released, which promoted the functional value method from a theoretical stage to a standardized practice. In 2021, the GEP accounting concepts and methodologies were included in the Framework for Ecosystem Accounting (SEEA-EA) by the United Nations Statistical Commission. Many works in the literature use the functional value method to calculate the value of ecosystem service or put forward eco-compensation standards [51,52,53].

Therefore, in addition to the equivalent-factor method, the functional value method (GEP accounting system) is also the current mainstream method; different valuation methods will be used for value accounting in the future, which may cause completely different compensation results.

(2)Uncertainty of validation

At present, one or two samples used for validation do not easily correspond to the value surface based on the annual average state, which is the spatiotemporal correspondence having poor performance with contingencies and uncertainties. In the future, on the one hand, the number of the sampling batches and coverage in space can be increased so that there is a better validation response in time and space; on the other hand, it may be necessary to use other methods, such as the InVEST model, to estimate water evapotranspiration to provide better experimental conditions for model validation.

## 5. Conclusions

Entropy curved surface is a creative method based on equivalent factors and MEP. A set of interregional, horizontal, ecological regulating service value, curved-surface models were constructed, an important improvement on the original method. The equivalent-factor method can be used to estimate ecosystem value quickly and effectively and can also be adjusted according to the actual situation. Maximum entropy production explains the flow features of the ecological function. In the peaks of ecological value, the value might be normally distributed with high probability and is in dynamic equilibrium. Therefore, the value is regarded as a tangible normal distribution surface. The expansion of the surface expresses the external output of value, and the eco-compensation relation becomes clear.

In the example of E’zhou City, the new method was used to calculate the eco-compensation standard. It found a rule for the ecological regulation function value of a macro morphological structure, simulated the transmission and diffusion of multiple values in interregional areas, solved the calculation problem of the horizontal ecological comprehensive compensation standard, and clarified the relation of “what, who, and how much should be compensated”. The new method improves the systematic and scientific characteristics of eco-compensation results and makes the model more consistent with the characteristics of the energy flow of an ecosystem. The analysis is programmed, standardized, and objectified with less subjective thought. It has better universality, streamlines steps, and markedly simplifies the logical process.

The results show that the ecological value of E’zhou City was 15.63 billion CNY, that of the E’cheng District was 5.224 billion CNY, that of the Huarong District was 5.123 billion CNY, and that of the Liangzi Lake District was 5.284 billion CNY. The unit area values of the three districts were 85,956 CNY/ha, 103,959 CNY/ha, and 106,548 CNY/ha, respectively. Little difference was observed in ecological value among the three districts, but the ecological value density of the Liangzi Lake District was the highest. Furthermore, ArcGIS and MATLAB applications were used as the main tools to numerically calculate the horizontal eco-compensation, and the final compensation results among the three districts of E’zhou were as follows: Liangzi Lake District should be compensated 114.40 million CNY, Huarong District should pay 69.90 million CNY, and E’cheng District should pay 44.50 million CNY.

This type of eco-compensation or payment for ecosystem services is likely to be faced anywhere in the world. No matter what compensation means are adopted, it is necessary to improve the scientific nature of eco-compensation standards and provide bases for compensation. This study demonstrated a novel use of MEP and provided empirical evidence to decision-makers in the field of economic values of ecosystem services and environmental management and sustainability. The method is valid at a city or county scale to solve the problem of transferring financial payments between horizontal interregional areas. The model studied in this paper and this kind of thinking are relevant to scholars in China or abroad.

## Figures and Tables

**Figure 1 entropy-25-01319-f001:**
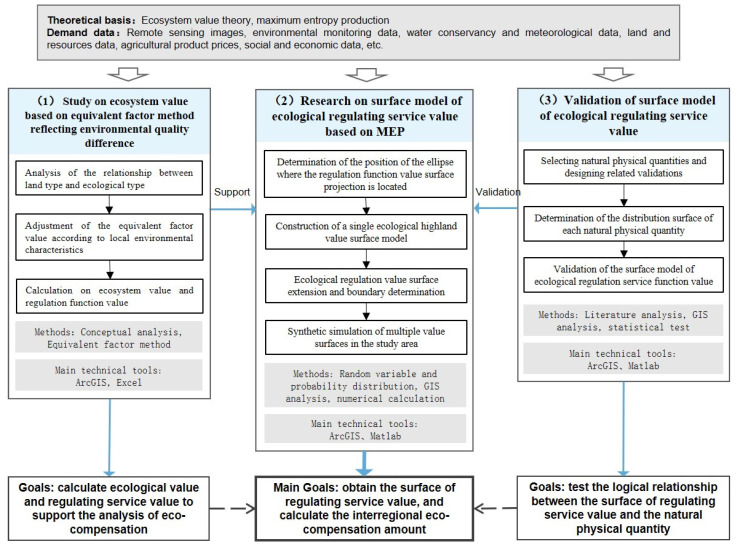
Research framework and steps.

**Figure 2 entropy-25-01319-f002:**
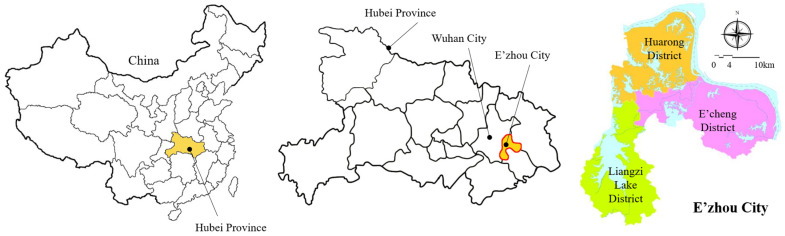
Study area: E’zhou City.

**Figure 3 entropy-25-01319-f003:**
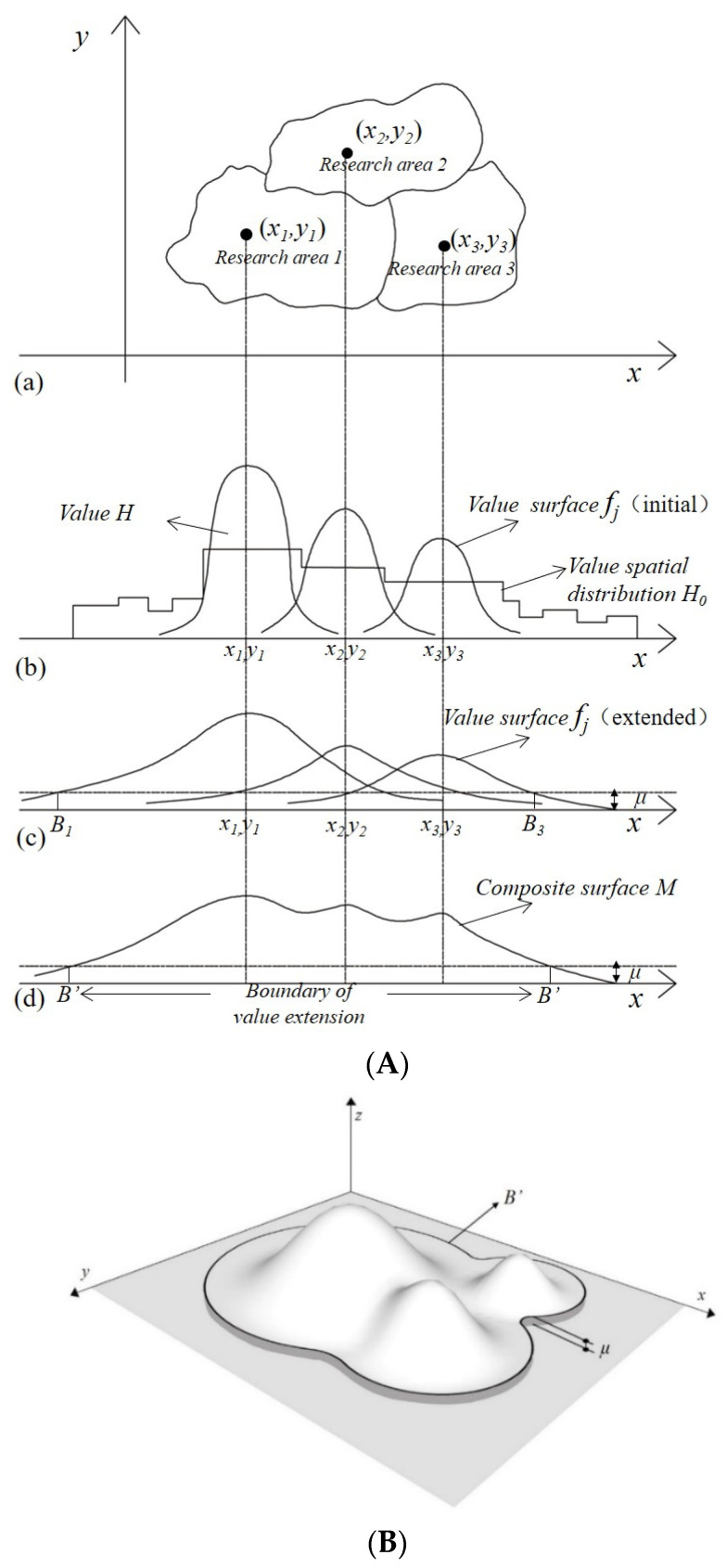
(**A**) The subfigure labels (**a**–**d**) illustrate that (**a**) the geometric center of the research area corresponding to the center of the ellipse, (**b**) the center of initial value surface corresponding to the center of the ellipse, (**c**) extension of a single surface, (**d**) synthesis of multiple surfaces. Two-dimensional section diagram of value surface showing that *H*_0_ is the natural distribution of the regulation value, and *H* is the regulation value of *H*_0_ after normalization and visualization. The total value of *f_j_* before and after expansion remained unchanged, and *μ* was the background value of ecological value per unit area within the research scope, such as the value of wasteland and bare land. (**B**) Three-dimensional diagram of the synthetic value surface. The *x*-*o*-*y* plane represents geographical coordinates, and *z* represents the regulating service value per unit area.

**Figure 4 entropy-25-01319-f004:**
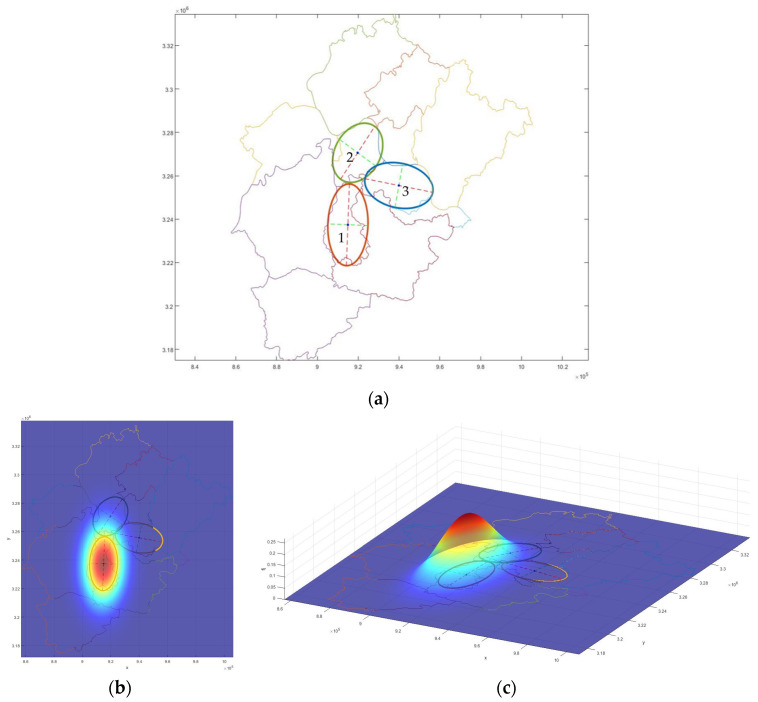
(**a**) Boundary of each district and fitting elliptical boundaries. (**b**) Top view and (**c**) isometric view of initial value-curved surface of gas regulation in Liangzi Lake District as an example.

**Figure 5 entropy-25-01319-f005:**
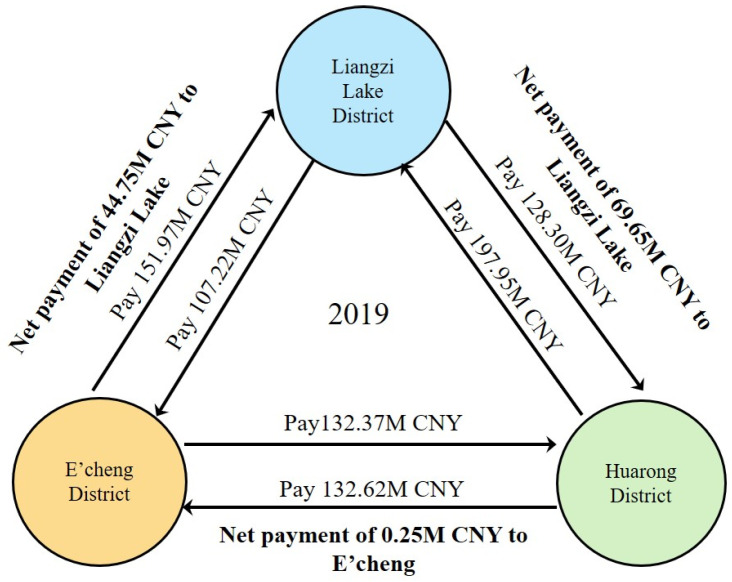
Relationship and net payments of eco-compensation among E’zhou’s three districts.

**Figure 6 entropy-25-01319-f006:**
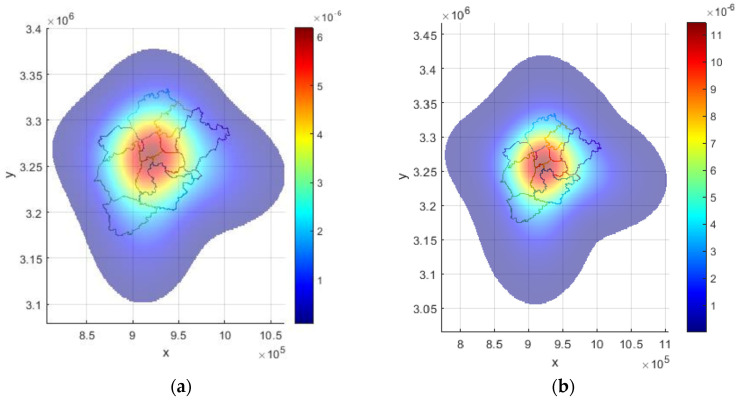
Synthetic curved surface of each regulating service value. Top view: (**a**) Gas regulation. (**b**) Climate regulation. (**c**) Environmental regulation. (**d**) Hydrological regulation. Isometric view: (**e**) Gas regulation. (**f**) Climate regulation. (**g**) Environmental regulation. (**h**) Hydrological regulation.

**Figure 7 entropy-25-01319-f007:**
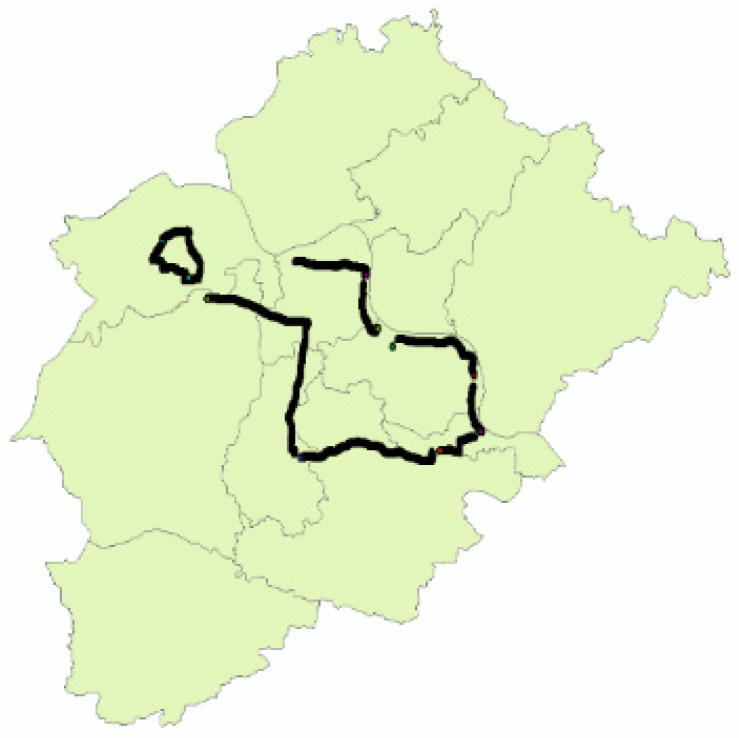
Sniffer4D with vehicle running track.

**Figure 8 entropy-25-01319-f008:**
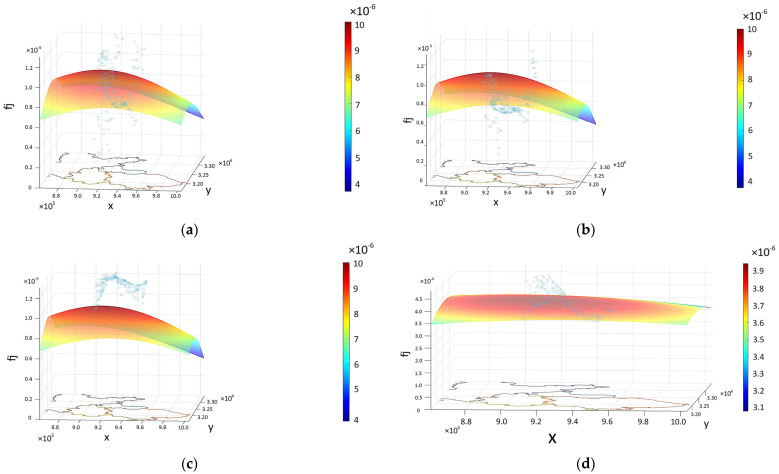
Physical quantity correlations between various value surfaces with the same corresponding coordinates in space. (**a**) PM_2.5_ and gas regulation. (**b**) PM_10_ and gas regulation. (**c**) VOC and gas regulation. (**d**) Humidity and climate regulation.

**Table 1 entropy-25-01319-t001:** Adjusted equivalent-factor table.

		Provisioning Services			Regulating Services				Supporting Services			Cultural Services	
Ecosystem Classification	Quality	Food Production	Raw Materials	Water Supply	Gas Regulation	Climate Regulation	Waste Treatment	Water Regulation	Soil Formation	Nutrient Cycling	Biodiversity	Cultural Recreation	Total
Dry land		0.85	0.40	0.02	0.67	0.36	0.10	0.27	1.03	0.12	0.13	0.06	4.01
Paddy field		1.36	0.09	−2.63	1.11	0.57	0.17	2.72	0.01	0.19	0.21	0.09	3.89
Broad-leaved forest	Low	0.15	0.33	0.17	1.09	3.25	0.97	2.37	1.33	0.10	1.21	0.53	11.48
	Medium	0.29	0.66	0.34	2.17	6.50	1.93	4.74	2.65	0.20	2.41	1.06	22.95
	High	0.44	0.99	0.51	3.26	9.75	2.90	7.11	3.98	0.30	3.62	1.59	34.43
Garden		0.75	0.21	−1.23	1.10	1.91	0.57	2.55	0.67	0.15	0.71	0.31	7.68
Shrub grass		0.38	0.56	0.31	1.97	5.21	1.72	3.82	2.40	0.18	2.18	0.96	19.69
Water area and wetland	I	0.88	0.86	4.48	3.28	6.22	6.22	24.23	3.99	0.31	13.60	8.17	72.25
	II	0.73	0.72	3.73	2.74	5.18	5.18	24.23	3.33	0.26	11.33	6.81	64.25
	III	0.61	0.60	3.11	2.28	4.32	4.32	24.23	2.77	0.22	9.44	5.68	57.58
	IV	0.51	0.50	2.59	1.90	3.60	3.60	24.23	2.31	0.18	7.87	4.73	52.02
	V	0.41	0.40	2.07	1.52	2.88	2.88	24.23	1.85	0.14	6.30	3.78	46.46
Desert		0.01	0.03	0.02	0.11	0.10	0.31	0.21	0.13	0.01	0.12	0.05	1.10
Bare land		0.00	0.00	0.00	0.02	0.00	0.10	0.03	0.02	0.00	0.02	0.01	0.20

**Table 2 entropy-25-01319-t002:** Adjustment coefficient of each value surface in the three districts.

	*μ* (CNY/km^2^)	Adjustment Coefficients *b* and *c*		
		Liangzi Lake	Huarong	E’cheng
Gas regulation	2115.927	2.26	2.70	2.41
Climate regulation	777.0786	2.36	2.83	2.58
Waste treatment	8714.648	1.84	1.82	1.64
Water regulation	3523.576	2.36	2.84	2.58

**Table 3 entropy-25-01319-t003:** Eco-compensation amounts between the three districts (unit: million CNY).

	L to H	L to E	H to L	H to E	E to L	E to H	L’s Income	H’s Income	E’s Income
Gas regulation	12.66	11.28	18.07	13.76	13.86	13.02	7.98	−6.14	−1.84
Climate regulation	15.54	14.94	25.93	17.61	20.38	15.91	15.82	−12.08	−3.74
Waste treatment	19.92	14.82	29.24	23.26	19.69	21.35	14.18	−11.22	−2.96
Water regulation	80.17	66.16	124.71	77.98	98.02	82.08	76.40	−40.44	−35.96
Total	128.30	107.22	197.95	132.62	151.97	132.37	114.40	−69.90	−44.50

Note: L, H, and E stand for Liangzi Lake, Huarong, and E’cheng districts, respectively.

**Table 4 entropy-25-01319-t004:** Comparison of the previous and new eco-compensation methods.

	Entropy Flat Surface (Previous)	Entropy Curved Surface (New)
Advantages	In calculation, key data such as the equivalent-factor table, land resource area, and GDP are needed, which Excel can calculate and is convenient for non-professional research personnel to obtain.	(1) The results of eco-compensation should be made more systematic and scientific, and the model should be more in line with the characteristics of ecosystem energy flow. (2) Analysis is programmed, standardized, and objectified with less subjectivity. (3) Its greater universality streamlines steps with no need to set up designated areas, markedly simplifying the logical process.
Disadvantages	(1) The model is idealized, which does not conform to the characteristics of energy flow and the dissipation structure of ecosystems and is not a normal distribution. (2) Determining the data of designated areas outside the research area is necessary. Data, such as GDP, administrative divisions, and various land resources, need to be determined, which is in great demand. (3) Compensation funds should be allocated according to GDP and ecological values.	The calculation operation is more difficult for administrative staff. ArcGIS and MATLAB software is needed for analysis and operation.

## Data Availability

Data available on request due to restrictions eg privacy or ethical.

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
