# Peer review of "Investigating a Method for a Horizontal Comprehensive Eco-Compensation Standard of Interregional Ecological Regulating Services"

_entropy, 2023, doi:10.3390/e25091319_

Round 1
Reviewer 1 Report (Previous Reviewer 1)
The research paper needs to be revised in the following aspects:
1) Please add a theoretical background, definitions of ecology, ecological regulations, and sustainability, also in reference to international experiences.
2) What are the practical implications of the conducted investigation? What are the added values for public policies? What could be the benefit for the regional societies?
Author Response
Please see the attachment.

Reviewer 2 Report (New Reviewer)
The manuscript undoubtedly touches on an important and timely topic of eco-compensation. Taking care of the environment is one of the most important issues today, so areas where standards may be exceeded are constantly monitored.
However, the compensation procedure is regulated by law, and any methods related to it should take into account the current regulations. It therefore seems reasonable to also refer to the legal and administrative requirements associated with eco-compensation.
In Model 1, the authors indicate 11 specific ecological functions and 8 ecosystem types (line 202). Please list these functions and types clearly (at least indicate them explicitly in Table 1).
Also indicate the limitations associated with the proposed method.
Author Response
Please see the attachment.

Reviewer 3 Report (New Reviewer)
The manuscript titled "Method for a horizontal comprehensive eco-compensation standard of interregional ecological regulation functions" presents an innovative approach to addressing the challenges associated with horizontal eco-compensation. The authors introduce the concept of the "Entropy Curved-Surface" method, a refined version of the previous "Entropy Flat Surface" model, to determine comprehensive multifactor compensation standards across different regions. The study aims to bridge the gap in existing horizontal eco-compensation strategies by providing a more accurate and ecologically sensitive solution.
The abstract succinctly highlights the primary motivations and objectives of the research, addressing the limitations of the previous model and the enhancements made in the proposed methodology. The research is grounded in real-world application, as demonstrated through the case study involving E’zhou City. This practical application lends credibility to the proposed approach and its potential utility in similar contexts. Nevertheless, some issues must be addressed in order this manuscript to be ready for publication.
In particular,
1. The goal of the manuscript is still unclear in terms of the methodology. The authors should address this issue.
2. The authors should use the template of the journal correctly. At this point there are many issues with the section's fonts, the bibliography, formulas' numberring etc.
3. There are many papers within bibliography in Chinese. This is a problem for a non Chinese person to check whether the used bibliography is the proper one! Use more references from studies published in English. This is an international journal and not a Chinese one.
4. Use passive voice within the manuscript.
5. The English is clear in general but it should need (still) a thorough revision.
6. The section "Results and Discussion" should be two separates sections. The separation of the "Results" and "Discussion" sections in a scientific manuscript is essential to maintain clarity, enhance reader comprehension, and uphold the integrity of the research process. This structural division allows for a distinct presentation of empirical findings and their interpretation. The "Results" section serves as a platform to objectively present raw data, statistical analyses, and empirical outcomes, free from subjective analysis or speculation. This separation fosters transparency, enabling readers to independently assess the robustness of the research and draw their own conclusions. On the other hand, the "Discussion" section permits researchers to critically analyze and interpret the significance of the results within the broader context of existing literature and theoretical frameworks. It provides an opportunity to elucidate underlying trends, patterns, and implications, while also addressing any discrepancies or limitations. By distinctively segregating these sections, the manuscript achieves a balanced presentation of data-driven observations and informed insights, ultimately enriching the overall quality and credibility of the research narrative. At this point, your manuscript is count 28 pages, while only the ection "Results and Discussion" counts 12. This is not balanced.
7. Enlarge as mauch as you can all the images. Some of them (e.g. fig.1) are not readable.
The English is clear in general but it should need (still) a thorough revision.
Round 2
Reviewer 3 Report (New Reviewer)
I would like to thank the authors for addressing my comments. I believe that the manuscript is ready to be published.
This manuscript is a resubmission of an earlier submission. The following is a list of the peer review reports and author responses from that submission.
Round 1
Reviewer 1 Report
Remarks and questions to be answered:
- What is the main research question in this study?
- Why is E’zhou City the subject of the study? What is so special about mentioned location?
- There is almost no information on interregional ecological regulation functions. Please explain as it is the topic of the research.
- Why is the Maximum Entropy Production (MEP) method best for the research?
- Which paper part could be interesting for international readers, if any?
Before potential publishing, all the questions should be answered in as detail as possible.
The quality of the English Language is acceptable.
Reviewer 2 Report
The authors proposed Entropy curved surface based on equivalent factor and MEP, as an improved version of Entropy flat surface to solve the problem of horizontal ecological compensation. The method is novel and the topic is meaningful. However, there are some concerns that need to be addressed.
1. The authors conclude "Theoretically, E’Cheng and Huarong District wouldpay a total of 114 million CNY to Liangzi Lake District. It is a significant improvement over theoriginal method". How to demonstrate it's an improvement. What is the result from the Entropy Flat Surface model. Does higher compensation mean better performance here?
2. The authors should explain better how the ecosystem value and regulation service value based on equivalent factor method support the modeling of Entropy curve surface.
3. The authors should give more detailed analysis results that made the authors choose to use the third case (bare land, wasteland) as the background mu.
4. The authors mentioned "the (second) verification result is improved compared with thefirst verification after the expansion of data range". How are these verification results in section 3.3 connected with eco-compensation results in section 3.2(3). Are results in section 3.2 use enough data range for modeling?
5. The authors mentioned the new Entropy Flat Surface needs "Data, such as GDP, administrative divisions and various land resources". These are also very critical data, though in great demand. Could these data bring Entropy Flat Surface method a more convincing result than Entropy Curve Surface? How could Entropy Curve Surface method also take advantage of such data?
Need extensive editing of English
Reviewer 3 Report
I found this paper incomprehensible. It is unclear what the goal was, how the method works and how the results are obtained. The scientific quality is very low and I don't think this can be published.
The English is readable but would need a thorough revision.